# Optimized Therapeutic ^177^Lu-Labeled PSMA-Targeted Ligands with Improved Pharmacokinetic Characteristics for Prostate Cancer

**DOI:** 10.3390/ph15121530

**Published:** 2022-12-09

**Authors:** Yitian Wu, Xiaojun Zhang, Xiaojiang Duan, Xing Yang, Feng Wang, Jinming Zhang

**Affiliations:** 1Department of Nuclear Medicine, Chinese PLA General Hospital, Beijing 100853, China; 2Department of Nuclear Medicine, Peking University First Hospital, Beijing 100034, China; 3Institute of Medical Technology, Peking University, Beijing 100191, China; 4Department of Nuclear Medicine, Nanjing First Hospital, Nanjing Medical University, Nanjing 210006, China

**Keywords:** prostate cancer, PSMA, SPECT, ^177^Lu, therapeutic

## Abstract

Clinical trials have shown the significant efficacy of [^177^Lu]Lu-PSMA-617 for treating prostate cancer. However, the pharmacokinetic characteristics and therapeutic performance of [^177^Lu]Lu-PSMA-617 still need further improvement to meet clinical expectations. The aim of this study was to evaluate the feasibility and therapeutic potential of three novel ^177^Lu-labeled ligands for the treatment of prostate cancer. The novel ligands were efficiently synthesized and radiolabeled with non-carrier added ^177^Lu; the radiochemical purity of the final products was determined by Radio-HPLC. The specific cell-binding affinity to PSMA was evaluated in vitro using prostate cancer cell lines 22Rv1and PC-3. Blood pharmacokinetic analysis, biodistribution experiments, small animal SPCET imaging and treatment experiments were performed on normal and tumor-bearing mice. Among all the novel ligands developed in this study, [^177^Lu]Lu-PSMA-Q showed the highest uptake in 22Rv1 cells, while there was almost no uptake in PC-3 cells. As the SPECT imaging tracer, [^177^Lu]Lu-PSMA-Q is highly specific in delineating PSMA-positive tumors, with a shorter clearance half-life and higher tumor-to-background ratio than [^177^Lu]Lu-PSMA-617. Biodistribution studies verified the SPECT imaging results. Furthermore, [^177^Lu]Lu-PSMA-Q serves well as an effective therapeutic ligand to suppress tumor growth and improve the survival rate of tumor-bearing mice. All the results strongly demonstrate that [^177^Lu]Lu-PSMA-Q is a PSMA-specific ligand with significant anti-tumor effect in preclinical models, and further clinical evaluation is worth conducting.

## 1. Introduction

Prostate cancer represents a significant public health problem and a heavy medical burden throughout the world. The incidence and cancer-related death of prostate cancer ranked first and second among male malignant tumors, respectively [1,2,3,4]. The options of therapeutic regimens are mainly based on the different clinical stages of patients [5,6]. Radical prostatectomy (RP) and radical radiotherapy are effective for early and localized PCa patients [7]. For patients with advanced or metastatic PCa, endocrine therapy, radiotherapy, chemotherapy and biotherapy are often applied, but the effectiveness of these available protocols is far from satisfactory; almost all patients will have disease progression in a short time [8,9,10,11,12]. Radionuclide therapy, as a novel therapeutic option that uses the biological effects of radiation emitted by radionuclides in the process of decay (α or β-) to inhibit or destroy pathological tissues, has attracted much attention in recent years. The commonly used radionuclides mainly include ^131^I, ^125^I, ^89^Sr, ^90^Y, ^177^Lu, ^223^Ra, etc. [13,14,15,16]. The radionuclides or the targeted compounds radiolabeled with these radionuclides can accumulate at the tumor lesion selectively, making the tumor subject to high-dose irradiation and thus causing it to lose the ability to proliferate and cause metabolic disorder, apoptosis or death, thereby achieving the purpose of treatment.

Radionuclide therapy has developed rapidly in the treatment of prostate cancer in the past two decades. In 2013, the FDA approved radium dichloride ^223^Ra (Xofigo injection) for the treatment of castration-resistant prostate cancer (CRPC) with bone metastases. Research showed that bone pain in patients treated with Xofigo has been relieved, and their survival period has been prolonged by 3.6 months. However, due to the limitations of the mode of action of Xofigo (simulating the effect of calcium and accelerating bone regeneration), it only showed an effect in patients with bone metastases [17,18]. In addition to the limited efficacy of Xofigo, some phase II and phase III trials indicated that concurrent treatment with radium-223 and abiraterone/prednisone or enzalutamide led to an increase in the incidence of fractures [19,20], leading to regulatory restrictions on combination therapy. ^89^Sr chloride (^89^Sr) served as a palliative treatment option for patients with bone metastases, but the high percentage of non-responders and the low complete response rate limited its clinical application [14]. PSMA-targeted ligands labeled with radionuclides specifically bind to PCa tumors, reaching the purpose of precise treatment at the molecular level [21,22]. ^177^Lu has become one of the most widely used therapeutic radionuclides due to its good performance and relatively simple labeling method [23,24,25,26]. [^177^Lu]Lu-PSMA-617 was approved by the US Food and Drug Administration (FDA) on 23 March 2022 due to its positive results in the phase III study.

In our previous studies, three novel PSMA-targeted ligands, namely PSMA-Q, PSMA-4PY and PSMA-BP have been selected due to their high affinity to PSMA, specific accumulation in PSMA+ tumors and better pharmacokinetic characteristics than PSMA-617, which indicate their therapeutic potential for PCa tumors [27,28,29]. In this study, PSMA-Q, PSMA-4PY and PSMA-BP labeled with ^177^Lu were prepared, and their therapeutic effects were evaluated through in vitro and in vivo experiments.

## 2. Results

### 2.1. Radiochemical Synthesis and Quality Control

The compounds PSMA-BP, PSMA-Q and PSMA-4PY were synthesized and purified by HPLC with a chemical purity of 98%; the MS analysis showed peaks of 940.44 (PSMA-BP), 991.99 (PSMA-4PY), 1042.50 (PSMA-Q) [M + H^+^] (Appendix A), respectively. Each of the radiotracers was prepared with a radiochemical purity (RCP) of more than 95% as analyzed by radio-HPLC, with retention times of 6.79 min ([^177^Lu]Lu-PSMA-BP^29^), 3.81 min ([^177^Lu]Lu-PSMA-4PY) and 5.11 min ([^177^Lu]Lu-PSMA-Q), respectively (shown in Figure 1). The molar activities (Am) were roughly calculated as 14.56 ± 3.33, 16.21 ± 5.11 and 18.42 ± 3.98 GBq/μmol.

### 2.2. Partition Coefficient and Stability

The log *p* values of [^177^Lu]Lu-PSMA-BP, [^177^Lu]Lu-PSMA-4PY and [^177^Lu]Lu-PSMA-Q were −2.31 ± 0.08 [29], −3.46 ± 0.10 and −3.29 ± 0.27, respectively. Among these, [^177^Lu]Lu-PSMA-Q showed the highest hydrophilicity. After incubation of each tracer in saline or 5% BSA at 37 °C for 2 h, a single peak was observed on the HPLC chromatogram, indicating good stability in the two systems in vitro within the tested time (shown in Figure 2).

### 2.3. In Vitro Cellular Studies

As human prostate cancer epithelial cell lines, 22Rv1 (mild PSMA+) and PC-3 (PSMA−) were used to evaluate the specific cell-binding affinity of the radiotracers. As shown in Figure 3, the uptake in 22Rv1 cells increased within the time tested, and all the novel tracers revealed a substantial high uptake in 22Rv1 cells; among these, [^177^Lu]Lu-PSMA-Q showed the highest uptake in 22Rv1 cells (3.65 ± 0.27 IA%/10^6^ at 24 h), followed by [^177^Lu]Lu-PSMA-BP (3.18 ± 0.30 IA%/10^6^) and [^177^Lu]Lu-PSMA-4PY (2.96 ± 0.47 IA%/10^6^). Uptake in 22Rv1 cells was significantly blocked by co-incubation with 2-PMPA. However, uptake of each novel tracer in PC-3 cells (lower than 0.5 IA%/10^6^ at 24 h) was much lower than that in 22Rv1 cells (*p* < 0.01), and could not be blocked by co-incubation with 2-PMPA. For comparison, the uptake of [^177^Lu]Lu-PSMA-617 in 22Rv1 cells was 3.41 ± 0.24 IA%/10^6^ at 24 h, which was similar to [^177^Lu]Lu-PSMA-Q (*p* = 0.22), but higher than those of [^177^Lu]Lu-PSMA-BP and [^177^Lu]Lu-PSMA-4PY. The percentages of internalization were 69.81 ± 1.01% ([^177^Lu]Lu-PSMA-Q), 64.47 ± 2.12% ([^177^Lu]Lu-PSMA-BP), 66.50 ± 2.33 ([^177^Lu]Lu-PSMA-4PY) and 64.62 ± 3.06 ([^177^Lu]Lu-PSMA-617), respectively.

### 2.4. Pharmacokinetics and Biodistribution Studies

Pharmacokinetics and biodistribution studies showed that all novel ligands could be rapidly eliminated from the blood, with clearance half-lives of 29.66 [29] ([^177^Lu]Lu-PSMA-BP), 21.27 ([^177^Lu]Lu-PSMA-4PY) and 24.30 min ([^177^Lu]Lu-PSMA-Q), respectively, which were all shorter than that of [^177^Lu]Lu-PSMA-617 (31.95 min) (shown in Figure 4).

As shown in Figure 5 and Table 1, [^177^Lu]Lu-PSMA-Q showed the highest uptake in the tumor at 24 h p.i. (3.87 ± 0.32 ID%/g), which was similar to that of [^177^Lu]Lu-PSMA-617 (3.74 ± 0.29 ID%/g, *p* = 0.63), both of which were significantly higher than those of [^177^Lu]Lu-PSMA-BP (2.98 ± 0.21 ID%/g) and [^177^Lu]Lu-PSMA-4PY (2.61 ± 0.49 ID%/g) (*p* < 0.05). Due to the fast clearance rate, the uptake of [^177^Lu]Lu-PSMA-R in the blood was lower than 0.50 ID%/g 24 h p.i. In normal organs, the highest uptake was observed in the kidney (1.21 ± 1.02 ID%/g for [^177^Lu]Lu-PSMA-BP, 1.64 ± 0.96 ID%/g for [^177^Lu]Lu-PSMA-4PY, 2.21 ± 0.87 ID%/g for [^177^Lu]Lu-PSMA-Q and 2.67 ± 1.02 ID%/g for [^177^Lu]Lu-PSMA-617), while other normal organs showed very low radioactivity accumulation and rapid clearance.

Due to the high uptake in the tumor and the fast clearance rate, the tumor-to-blood (T/B), tumor-to-muscle (T/M) and tumor-to-kidney ratios were 17.59, 48.38 and 1.75, which were higher than those of [^177^Lu]Lu-PSMA-617(15.58 for T/B, 46.75 for T/M and 1.40 for T/K), [^177^Lu]Lu-PSMA-BP (9.61 for T/B, 29.80 for T/M and 1.34 for T/K), and [^177^Lu]Lu-PSMA-4PY (6.07 for T/B, 29.00 for T/M and 1.59 for T/K).

### 2.5. Small-Animal SPECT

SPECT images of 22Rv1 tumor-bearing mice at 24 h p.i. confirmed high uptake in tumors, medium uptake in kidneys, and very low uptake in other normal organs of [^177^Lu]Lu-PSMA-R and [^177^Lu]Lu-PSMA-617 (Figure 6). [^177^Lu]Lu-PSMA-Q showed the highest tumor-to-muscle ratio of 16.77 ± 3.27, followed by [^177^Lu]Lu-PSMA-617 (14.51 ± 4.64), [^177^Lu]Lu-PSMA-BP (10.21 ± 5.66) and [^177^Lu]Lu-PSMA-4PY (7.54 ± 3.85).

### 2.6. Therapy Study

To evaluate antitumor efficacy, a pilot experiment was conducted using 5 groups of 22Rv1 tumor-bearing nude mice, injected with 74 MBq of [^177^Lu]Lu-PSMA-BP, [^177^Lu]Lu-PSMA-4PY, [^177^Lu]Lu-PSMA-Q, [^177^Lu]Lu-PSMA-617 or saline (*n* = 10/group). Tumor volume and body weight for each group are shown in Figure 7.

The injection day was defined as Day 0, and day the mice die or reach any endpoint criterion was defined as the termination day. Tumor-bearing mice without any of the above conditions were continuously observed until Day 32.

The results showed that the weight of the mice in each experimental group decreased slightly in the short term after the administration of ligands and then remained relatively stable. At Day 32, the weight of mice in the [^177^Lu]Lu-PSMA-BP and [^177^Lu]Lu-PSMA-4PY groups decreased compared with before treatment (*p* < 0.05), while there were no statistical differences in the weight of [^177^Lu]Lu-PSMA-Q and [^177^Lu]Lu-PSMA-617 groups before and after treatment (*p* > 0.05). The weight of the control group decreased significantly within the experimental period (*p* < 0.05).

Eight days post injection, the tumor volume of the [^177^Lu]Lu-PSMA-Q and [^177^Lu]Lu-PSMA-617 groups was already significantly lower than that of the control group (*p* < 0.05). The ratio of tumor volume at Day 8 to tumor volume at Day 0 (vt/v0) was 3.53 ± 1.14, 1.49 ± 0.33 and 1.34 ± 0.28 for control, [^177^Lu]Lu-PSMA-Q and [^177^Lu]Lu-PSMA-617 groups, respectively. At Day 11, in the [^177^Lu]Lu-PSMA-4PY group, the tumor volume was significantly smaller than that in the control group (*p* < 0.05); vt/v0 was 6.81 ± 1.40 for the control group and 3.24 ± 0.85 for [^177^Lu]Lu-PSMA-4PY group at Day 11. On the day of the end of the study, there was no significant difference in tumor volume between the [^177^Lu]Lu-PSMA-Q and [^177^Lu]Lu-PSMA-617 groups (*p* = 0.12), with vt/v0 of 2.10 ± 0.68 ([^177^Lu]Lu-PSMA-Q) and 2.75 ± 1.04 ([^177^Lu]Lu-PSMA-617), which were significantly smaller than that in [^177^Lu]Lu-PSMA-4PY (vt/v0 = 8.24 ± 1.75) and control groups (vt/v0 = 20.77). The tumor growth in the BP group was also slower than in the control group, but there was no significant difference throughout the whole experimental period (*p* > 0.05) (shown in Figure 8).

On the day at the end of the entire treatment experiment (Day 32), the survival rates were 80% ([^177^Lu]Lu-PSMA-Q), 60% ([^177^Lu]Lu-PSMA-617), 50% ([^177^Lu]Lu-PSMA-4PY), 20% ([^177^Lu]Lu-PSMA-BP) and 0 (control), respectively. The survival time of the mice in each treatment group was significantly longer than that of the control group (*p* < 0.01) (shown in Figure 9).

### 2.7. Radiotoxicity

The mice in the experimental groups were injected with 74 MBq of [^177^Lu]Lu-PSMA-R or [^177^Lu]Lu-PSMA-617, and there was no death within the observation period. There were no significant differences in diet, excretion, activity, mental state, skin condition, and weight change between the experimental group and the control group (*p* > 0.05). No significant differences were observed in the size, color, shape, and texture of the organs among the five groups. The kidney weights of the mice in each group were 0.191 ± 0.03 g ([^177^Lu]Lu-PSMA-BP), 0.192 ± 0.02 g ([^177^Lu]Lu-PSMA-4PY), 0.186 ± 0.02 g ([^177^Lu]Lu-PSMA-Q), 0.188 ± 0.03 g ([^177^Lu]Lu-PSMA-617) and 0.185 ± 0.02 g (control), respectively. There were no significant differences in the results of the HE staining of the sections of the main organs among the five groups; moreover, no obvious toxic damage, such as necrosis and fibrosis, was observed under the microscope. The blood routine indexes of mice in each group remained within the normal reference value range on the 14th and 28th days after the injection of radioactive drugs (shown in Figure 10).

## 3. Discussion

Prostate cancer is one of the most highly prevalent malignancies in men. By 2020, prostate cancer had become the most commonly diagnosed cancer among men in 112 countries [30]. Since the development of advanced prostate cancer is mainly driven by the androgen signaling pathway, hormone therapy is the most commonly used treatment for patients with prostate cancer who have lost the opportunity for radical surgery and radical radiotherapy. In the early stage of anti-androgen therapy, prostate cancer is most effectively controlled; however, with extension of the treatment period, almost all patients experience progression that is difficult to suppress with hormone therapy [8,9]. During the course of anti-androgen therapy, testosterone remains at depleted levels, and prostate cancer progression within this period is referred to as castration-resistant prostate cancer (CPRC) [31]. Currently, docetaxel, a paclitaxel antineoplastic drug, is mostly used as a first-line treatment option for CPRC. However, after multicycle chemotherapy, most patients experience significant toxic side effects and develop drug resistance, and the growth and invasion rate of tumors increases significantly again [32,33,34]. The new generation of antiandrogen drugs, abiraterone and enzalutamide, have improved progression-free and overall survival in patients with CRPC to some extent, but the improvement of the overall prognosis was limited. In addition, studies found that resistance to either abiraterone or enzalutamide was always accompanied by resistance to another new generation anti-androgen drug and docetaxel [35,36,37,38]. On the basis of the above points, the treatment of CRPC remains an important and difficult problem to be solved.

Prostate specific membrane antigen (PSMA), which is specifically overexpressed on almost all prostate cancer cell membranes, is considered to be an ideal target for the diagnosis and treatment of prostate cancer; PSMA-mediated radionuclide therapy (PRRT) has also been a research hotspot in recent years. A combination of β-rays-emitted radionuclides and PSMA-targeted ligands can bind to the surface of PCa cells and further internalize into cells, thus exerting the effect of killing the tumor. ^177^Lu is the most commonly used radionuclide for radiotherapy; it emits β-rays of moderate energy (Emax = 0.5 MeV), with a tissue penetration range of approximately 2 mm and a physical half-life of 6.7 days, while the γ-photons emitted by ^177^Lu can also be used for SPECT imaging to monitor the curative effect before and after treatment. Studies have confirmed the safety and effectiveness of ^177^Lu-labeled PSMA ligands such as [^177^Lu]Lu-PSMA-617 and [^177^Lu]Lu-PSMA-I & T in the treatment of prostate cancer, but their pharmacokinetic properties for PCa treatment, among others, still need to be improved [39,40]. In this study, we evaluated and compared the therapeutic potential of the PSMA ligands [^177^Lu]Lu-PSMA-Q, [^177^Lu]Lu-PSMA-BP and [^177^Lu]Lu-PSMA-4PY, which exhibited high affinity and excellent pharmacokinetics in our previous studies. First, we verified the stability of these ligands and measured their hydrophilicity through in vitro experiments. [^177^Lu]Lu-PSMA-4PY showed the highest hydrophilicity, followed by [^177^Lu]Lu-PSMA-Q and [^177^Lu]Lu-PSMA-BP. The stronger hydrophilicity facilitated the clearance of the ligands from blood, normal tissues, and organs, reducing the possibility of toxic reactions. Subsequent animal toxicity experiments also proved their safety again. After euthanasia and autopsy of the mice, there was no significant difference in the size, color, texture, and weight of the main organs of the mice in either the experiment or control group. The results of organ HE staining showed that there were no abnormalities in organs among five groups, especially in the kidney and salivary glands, which were vulnerable to radiation. No abnormality was found in the blood routine tests in each group.

The purpose of radionuclide endoradiotherapy is to maximize the absorbed dose of radionuclide by tumors, while ensuring that uptake of other normal tissues is within a reasonable range. In the in vitro cellular uptake assay, all novel ligands were specifically accumulated in 22Rv1 cells with PSMA-positive expression, among which [^177^Lu]Lu-PSMA-Q showed the highest uptake, followed by [^177^Lu]Lu-PSMA-BP and [^177^Lu]Lu-PSMA-4PY; there was almost no uptake in PC-3 cells. Further biodistribution and SPECT imaging results showed that the three novel ligands could be specifically uptaken by 22Rv1 tumors. The uptake in the tumors was higher than in other normal organs. The salivary glands, which were vulnerable to radiation damage, showed low uptake of all ligands, with uptake values less than 0.5 %ID/g 24 h p.i. in the biodistribution experiment. The tumor, kidney, and bladder were clearly visualized on SPECT images at 24 h p.i., and other background tissues showed extremely low uptake. The tumor-to-muscle ratios of [^177^Lu]Lu-PSMA-Q, [^177^Lu]Lu-PSMA-BP, [^177^Lu]Lu-PSMA-4PY and [^177^Lu]Lu-PSMA-617 were 16.77 ± 3.27, 10.21 ± 5.66, 7.54 ± 3.85 and 14.51 ± 4.64, respectively. The result demonstrates that the accumulation of the ligands within the tumor was persistent; meanwhile, they were rapidly washed out from normal organs through urinary metabolism.

In the therapy study, 22Rv1 tumor-bearing mice were treated with 74 MBq of [^177^Lu]Lu-PSMA-Q, [^177^Lu]Lu-PSMA-BP and [^177^Lu]Lu-PSMA-4PY. The results showed that the body weight of mice in the treatment and control groups decreased to a certain extent in a short period. This was due to the tumor itself being in a multiplicative stage of growth; cell growth was vigorous and a lot of nutrients were needed. However, the weight of tumor-bearing mice in the Q group gradually stabilized around 8 d and increased slightly around 20 d after therapy. The weight of mice in the 617 group decreased slightly and began to stabilize around 14 d p.i. The weight of the BP and 4PY groups continuously decreased; however, the decrease was smaller than that of the control group, indicating that tumor cell growth activity in the treatment groups had been inhibited to some extent. However, mice in the control group continued to lose weight, and symptoms similar to cachexia, such as poor mental status, reduced mobility, and reduced food intake, appear in later stages. At the end of the treatment experiment, the weight of the mice in every treatment group was higher than that of the control group. In terms of tumor growth, compared with the control group, the tumor growth rate of all treatment groups was significantly slower. [^177^Lu]Lu-PSMA-Q had a significant inhibitory effect on tumor growth, which was equivalent to [^177^Lu]Lu-PSMA-617. At the end of the treatment experiment, there was no significant difference in the average tumor volume between the Q group and the 617 group (*p* > 0.05), and both were smaller than that of control group (*p* < 0.05). While [^177^Lu]Lu-PSMA-4PY and [^177^Lu]Lu-PSMA-BP showed slightly weaker inhibitory effects on the tumors, the tumor growth rate was still lower than that of the control group. In terms of the survival time of the mice in each group, 22 days after the start of the treatment experiment, all tumor-bearing mice in the control group died or reached the endpoint criteria. The survival rates of [^177^Lu]Lu-PSMA-Q, [^177^Lu]Lu-PSMA-BP, [^177^Lu]Lu-PSMA-4PY, and [^177^Lu]Lu-PSMA-617 at the termination were 80%, 20%, 50% and 60%. The survival rate of each group in the experimental group was significantly higher than that of the control group. Within the four treatment groups, survival rates were not statistically different among Q, 617 and 4PY groups (*p* > 0.05); however, the Q and 617 groups were higher than the BP group (*p* < 0.05), and there was no difference between the 4PY and BP groups (*p* > 0.05).

In terms of the effect of structural modifications on therapeutic efficacy, incorporation of the 4-pyridyl, 3-quinoline, or biphenyl moiety enhances the hydrophilicity of the ligand, especially the 4-pyridyl. With the guarantee of the affinity of the ligands, a higher hydrophilicity indicates a faster blood clearance rate and a higher tumor-to-background ratio, which may be beneficial to tumor treatment efficacy and reduce toxic side effects in accordance with our expectation. However, the relatively low binding affinity and excessive hydrophilicity of [^177^Lu]Lu-PSMA-4PY lead to low uptake in tumors and rapid clearance of [^177^Lu]Lu-PSMA-4PY, not only from blood and normal organs, but also from the tumor, resulting in undesirable therapeutic efficacy. Compared with 4-pyridine, the incorporation of 3-quinoline affects the affinity of the ligand to a small extent, which is coupled with its moderate hydrophilicity, resulting in an ideal outcome. In addition to the above, actually, because of the complexity of the in vivo environment, the results of in vivo therapeutic studies are difficult to predict purely based on the characteristics of ligands and the results of in vitro studies. Regardless of the modification of the structure, the results of in vivo therapeutic studies are the ultimate reflection of whether the modification meets initial expectations.

However, there are still limitations in this study. First, the inhibitory effect of ligands on tumors in this study is lower than that reported in some of the literature. The reasons for this are, on the one hand, the large size of the tumor used for the evaluation of the efficacy in this study compared with that reported in the literature (100 mm^3^ vs. 20 mm^3^), and the fact that larger tumors are somewhat weakened by the therapeutic effect of radionuclides due to possible necrosis in their centers and the difficulty of penetration of radiation; on the other hand, the 22Rv1 cells have a lower level of PSMA expression than LNCaP and PC-3 PIP cells. Second, in the evaluation of drug toxicity, the observation period was relatively short; only the absence of acute toxic effects of the drugs could be confirmed. Third, multiple cycles of treatment are commonly used in clinical patients; however, in this study, only a single dose was administered, so the safety and efficacy of multiple treatments still need further study. Fourth, the absorbed dose of the ligands in various organs should be estimated to ensure that the dose in normal tissues and organs is within an acceptable range. However, as a preclinical study, the basic experimental results provided by this study confirmed the potential of [^177^Lu]Lu-PSMA-Q for prostate cancer treatment; this was sufficient to support clinical transformation and large-scale prospective studies to further validate its effectiveness, and to ensure that the effectiveness is not inferior to that of [^177^Lu]Lu-PSMA-617, while producing less radiation damage and toxic effects.

## 4. Materials and Methods

All chemicals, reagents, and solvents for the synthesis and analysis were of analytical grade (all purchased from Maclin Biochemical Technology Co., Ltd. (Shanghai, China)). ^177^LuCl^3^ solution was purchased from ITG Isotope Technologies Garching GmbH (München, Germany). All animal studies were performed according to a protocol approved by the Animal Care and Use Committee of Chinese PLA General Hospital.

### 4.1. Chemical Synthesis, Radiolabeling and Quality Control

The chemical synthesis of novel precursors and their radiolabeling procedures have been reported previously [29] (briefly shown in Figure 1; the details of compound preparation were described in Appendix A).

Radiochemical purity was determined via radio-high-performance liquid chromatography (HPLC) (Waters 515) equipped with a γ-detector (BIOSCAN Flow-Count, NaI scintillation detector). HPLC conditions: Waters Nova-Park C-18 column (3.9 × 150 mm, 5 µm), UV = 254 nm, acetonitrile/H_2_O (0.4% phosphoric acid (isocratic elution) = 25/75, 1 mL/min.

### 4.2. Partition Coefficient

The octanol-water partition coefficient was determined by adding [^177^Lu]Lu-PSMA-R (0.37 MBq in 100 μL), 2.9 mL of phosphate-buffered saline (0.1 M, pH 7.4) and 3 mL of octanol to a tube. The mixture was vortexed for 2 min and centrifuged (3000 rpm × 5 min). Then, 3 samples (1 mL) from each phase were collected and the radioactivity was measured by a γ-counter (Wallac 2480 Wizard, MA), respectively. The experiment was carried out in triplicate and repeated 3 times. Log *p* value was calculated using log *p* = log (CPM for octanol/CPM for water) and reported as Log *p* ± SD.

### 4.3. In Vitro Stability

The in vitro stabilities of [^177^Lu]Lu-PSMA-R were determined by measuring radiochemical purity (RCP) in saline and 5% bovine serum albumin (BSA), at 37 °C at different time intervals (1 h, 4 h, 12 h, 24 h, 36 h), using radio-HPLC.

### 4.4. Cell Lines and Culture Condition

Cell lines were kindly provided by the stem cell bank at the Chinese Academy of Sciences. The human prostate cancer cell lines 22Rv1 (mild PSMA+) and PC3 (PSMA-) were cultured in RPMI 1640 medium (Gibco, New York, NY, USA) and supplemented with 10% Fetal Bovine Serum (FBS, Gibco) and 1% penicillin-streptomycin (Gibco) in a humidified incubator at 37 °C under 5% CO_2_. Each cell line was passaged to around 80–90% confluence, and harvested using trypsin-ethylenediaminetetraacetic acid (trypsin-ETDA; 0.25% trypsin, 0.02% ETDA, all from Gibco).

### 4.5. Cell Binding and Internalization Studies

Cells were seeded in 24-well plates (1 × 10^5^ cells in 1 mL medium/well) and incubated at 37 °C/5% CO_2_ overnight; the medium was changed 2 h before the experiment. Each well was then added with ^177^Lu-PSMA-R (0.074 MBq in 100 μL saline) and incubated at 37 °C/5% CO_2_ for 24 h. After incubation, the medium was removed and the cells were washed twice with ice-cold phosphate-buffered saline (0.5 mL) at each time point; finally, the cells were lysed by NaOH (0.5 M, 0.5 mL). 2-(phosphonomethyl) pentane-1,5-dioic acid (2-PMPA) solution (2 µg/well) was used for the blocking experiment. For the internalization experiment, cells were incubated with glycine-HCL in PBS (50 nM, PH 2.8, 0.5 mL) for 5 min at room temperature, then washed twice with ice-cold PBS (0.5 mL) and lysed by NaOH (0.5 M, 0.5 mL). The radioactivity of cells was measured using a γ-counter. The studies were performed in triplicate and the result was reported as percentage injected activity (%IA)/10^6^ cells.

### 4.6. Pharmacokinetics

An injection of [^177^Lu]Lu-PSMA-R or [^177^Lu]Lu-PSMA-617 (7.4 MBq in 150 μL) was administered intravenously via tail into BALB/c male mice (4 groups, 3 mice/group), and a blood sample (5 μL) was taken from mouse orbit at 2, 5, 15, 30, 45, 60 and 120 min post injection. The radioactivity of the blood samples was measured by a γ-counter and the pharmacokinetics data were calculated using the WinNonlin 8.1 software (Certara, Princeton, NJ, USA).

### 4.7. Tumor Models

BALB/c nude mice (male, 4–6 weeks, 18–22 g) were purchased from Vital River Laboratory Animal Technology Co., Ltd. (Beijing, China). Each mouse was inoculated subcutaneously with PSMA-positive 22Rv1 or PSMA-negative PC-3 cells (1 × 10^7^ cells/mL, 0.2 mL) on the left flank; when the tumor had grown up to 80–150 mm^3^, biodistribution, SPECT imaging and radiotherapy studies were conducted.

### 4.8. Biodistribution

The 22Rv1 tumor-bearing mice were injected with [^177^Lu]Lu-PSMA-R or [^177^Lu]Lu-PSMA-617 (4 groups, *n* = 3/group, 0.74 MBq in 150 μL per mouse). Mice were sacrificed at 24 h p.i., tumors and interested organs were harvested, weighed and measured by a γ-counter, and the results were expressed as the percentage of injected dose per gram (ID%/g).

### 4.9. Small-Animal SPECT

Mice with subcutaneous 22Rv1 tumors (4 groups, *n* = 3/group) were injected intravenously with 11.1 Mbq [^177^Lu]Lu-PSMA-R or [^177^Lu]Lu-PSMA-617. 24 h after injection, the mice were scanned for 20 min using SPECT (SimensSymbia T6). CPM for ROI over tumor and muscle were measured.

### 4.10. Therapy Study

Mice with PSMA (+) 22Rv1 flank tumor models (4 groups, *n* = 10/group) were used to determine the antitumor efficacy of [^177^Lu]Lu-PSMA-R and [^177^Lu]Lu-PSMA-617. The mice were injected intravenously with [^177^Lu]Lu-PSMA-R or [^177^Lu]Lu-PSMA-617 (74 MBq, 200 μL), respectively. Then, the body weight of mice and tumor volume were measured every 2 days. The tumor volume was determined using V = width^2^ × length/2. Endpoint criteria were defined as weight loss ≥15%, tumor volume >1800 mm^3^, active ulceration of the tumor or abnormal behaviors.

### 4.11. Radiotoxicity

Healthy ICR male mice were injected with [^177^Lu]Lu-PSMA-R or [^177^Lu]Lu-PSMA-617 (74 MBq in 150 μL) or saline (150 μL) for the radiotoxicity evaluation (5 groups, *n* = 15/group). At 14 d post injection, venous blood was drawn from the tail vein for the routine blood test; subsequently, 5 of the mice in each group were euthanized and the main organs were harvested and fixed in 4% formalin in PBS. Organ samples were trimmed and embedded in paraffin. Then, histological sections were prepared, stained with hematoxylin-eosin and assessed under a light microscope. At 28 d post injection, the routine blood test was carried out again in the remaining mice in each group.

### 4.12. Statistical Analysis

All quantitative data were expressed as mean ± SD. The normality of the data was assessed by the Shapiro-Wilk test, followed by the two-tailed Student’s *t*-test or the Mann–Whitney U test. A *p* value of <0.05 was considered statistically significant. Statistical analyses were performed using SPSS software 22.0 (IBM Corp., Armonk, NY, USA) and Prism 8 software (GraphPad Software, San Diego, CA, USA).

## 5. Conclusions

In this study, we evaluated the radiotherapy efficacy of three novel ^177^Lu-labeled ligands selected from our previous study. [^177^Lu]Lu-PSMA-Q exhibited the highest affinity and excellent pharmacokinetic characteristics, resulting in a satisfactory tumor inhibition effect which was not inferior to [^177^Lu]Lu-PSMA-617 and is worth further study.

## Data Availability

Data is contained within the article and Appendix A.

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
