# Peer review of "Optimized Therapeutic 177Lu-Labeled PSMA-Targeted Ligands with Improved Pharmacokinetic Characteristics for Prostate Cancer"

_pharmaceuticals, 2022, doi:10.3390/ph15121530_

Round 1
Reviewer 1 Report
In this paper, the authors described the pharmacokinetic characteristics and therapeutic performance of a series of [177Lu]Lu-PSMA compounds. Among these compounds, the [177Lu]Lu-PSMA-Q was identified as the most effective therapeutic radiopharmaceuticals to suppress tumor growth and improve the survival rate of tumor-bearing mice. However, since [177Lu]Lu-PSMA-617 has been approved for clinical use, the concept of using this type of therapeutical agent is not new. It is suggested that the authors could have more discussion about the effects of structural differences of PSMA analogs on therapeutic outcomes. Overall, the manuscript is well-written but some important issues should be addressed. It could be considered for publication in Pharmaceuticals after the revision.
1. Page 2, line 61, the suppliers for the materials should be indicated.
2. Page 2, in “chemical synthesis, radiolabeling and quality control”, the details of the compound preparation are missing.
3. Page 2, in “chemical synthesis, radiolabeling and quality control”, many informations about the HPLC method are missing, e.g. brand and type of HPLC column, eluting conditions, etc.
4. Page 6, figure 1 & page 8, figure 4, the authors showed some data of [177Lu]Lu-PSMA-BP, which have published in their previous paper, J. Radioanal. Nucl. Chem. (2022) 331:2705–2717. The authors should pay attention to copyright and self-plagiarism issues. At least, it should have an appropriate citation.
5. Page 15, line 328, damage to salivary glands is a known side effect of PSMA-based radiation therapy. From the PK point of view, the radiolabeled-PSMAs may reach the highest level of uptake by the salivary glands within the first few hours after injection. The radiation damage at early time points should be considered. The biodistribution results at 24 h may not reflect this issue.
6. From the structural point of view, could the authors explain why [177Lu]Lu-PSMA-Q shows the best therapeutic performance? In other words, why is the quinoline substituent important?
Reviewer 2 Report
This is a well-designed study by the authors and indeed newer PSMA-ligands with improved pharmacokinetics for therapy are required to minimize adverse effects due to accumulation in normal organs. The title and conclusions need to be redone as we are not having improved pharmacokinetics to the existing ones, but just newer ligands with not inferior characteristics. The English requires editing by a native speaker.
Reviewer 3 Report
the manuscript titled Optimized Therapeutic 177Lu-labeled PSMA-targeted Radio- 2 pharmaceuticals with Improved Pharmacokinetic Characteris- 3 tics for Prostate Cancer, has many salient features scientifically as well as presentation-wise.
the work can work as model article for any theranostic approach studies. surely its a good read with detail informations. However, I am not convinced with a few sections that the author can address to improve the article quality. 1) did the tumor bearing mice were subjected for CT or MRI to obtain an actual correlation with the PET, in other words how do we know the tumor uptake is specific. 2) With respect to therapeutic effects, did the authors consider studying the drug clearance timeline if yes, that can be shown in any figure. 3) can the control mice biodistribution studies be presented to the significance. or any contralateral tissue analysis as control in biodistribution. 4) the color bar is missing in the spect images. 5) what could be the contrast seen outside the mice shape lines. 6) Please provide clear high resolution spect images coregistered images with CT or MRI for better understanding.Author Response
Please see the attachment.

Reviewer 4 Report
The manuscript submitted for evaluation deals with the current topic - prostate-specific membrane antigen in the therapy of prostate cancer. In the manuscript, 4 newly prepared PSMA derivatives labeled with 177Lu are studied. Both pharmacokinetics and biodistribution of the studied compounds are described in the manuscript. The data are sufficiently experimentally supported and are presented clearly and comprehensibly.
I have a few questions and comments about the manuscript:
1. The authors report that they used carrier-free 177Lu for labeling. A more correct term used in nuclear chemistry is non carrier added - n.c.a. For 177Lu, it is necessary to indicate the source of the radionuclide, because many producers have a problem with the preparation of 177Lu n.c.a. Could you specify it in the text?
2. In the paragraph dedicated to quality control, the authors state that they used a gamma detector for HPLC radiodetection. Here it is necessary to specify whether it was a NaI(Tl) - scintillation detector or they really used HPGe detection. It is necessary to specify detection parameters such as energy discrimination and energy window, even collimator when it was used and detection cell parameters. Also, the text on HPLC does not indicate whether gradient or isocratic techniques were used. And if analytical HPLC was also used for preparative separation (line 153). This needs to be added.
3. The authors state in the text that they prepared four 177Lu labeled PSMA derivatives, stating that their radiochemical yield was 95%. However, the total or specific activity of the prepared PSMA derivatives is not reported here. It is important to add here.
4. In the graphs in Figures 7, 8 and 9, the control group is terminated after approximately 23-25 days, while the rest after 32 days. It means that the control group outlasted 23-25th day? Can you comment on that?
I would also like to point out the comment on the introduction, where on lines 39-42 it is written about the therapeutic use of radionuclides and a mixture of radionuclides is mentioned here. A classic example established in practice for a long time is precisely 131I. Iodine-125 currently has no use other than brachytherapy, and even that is more of a historical matter. Palliative therapy with 89Sr and 223Ra is worth mentioning, but the benefit of Xofigo, in particular, has been discussed a lot recently, and there were rather complications in connection with Enzalutamide and Abiraterone, which led the EMA to restrictions and stricter indication criteria. This fact needs to be taken into account. From the point of view of PRRRT and prostate therapy, the benefit of both 177Lu and 90Y is indisputable.
Chromatograms on Fig.1. should be added in better quality and resolution.
Line 226 - should be indented.
In supplemetary file 1H NMR and 13C NMR data as well as FT-IR spectra should be added, beacause the mentioned structures are newly described. Also MS data should be described properly. This is necessary to pay an attention.
The submitted manuscript is written in an interesting and high-quality manner. The presented work is recent and therefore I recommend the manuscript for acceptance after minor corrections.
Round 2
Reviewer 3 Report
the revised manuscipt has been improved.